# Satellite Tracking of Head-Started Juvenile Green Turtles (*Chelonia mydas*) Reveals Release Effects and an Ontogenetic Shift

**DOI:** 10.3390/ani13071218

**Published:** 2023-03-31

**Authors:** Nicole Barbour, Helen Bailey, William F. Fagan, Walter Mustin, Vandanaa Baboolal, Francesca Casella, Tony Candela, Philippe Gaspar, Sean Williamson, Emily Turla, George L. Shillinger

**Affiliations:** 1Department of Environmental Biology, SUNY College of Environmental and Forest Sciences, Syracuse, NY 13210, USA; 2Chesapeake Biological Laboratory, University of Maryland Center for Environmental Science, Solomons, MD 20688, USA; 3Department of Biology, University of Maryland, College Park, MD 20742, USA; 4Upwell, Monterey, CA 93940, USAgeorge@upwell.org (G.L.S.); 5Cayman Turtle Conservation and Education Centre, Grand Cayman 1303, Cayman Islands; 6Mercator Ocean International, 31400 Toulouse, France; 7School of Biological Sciences, Monash University, Clayton 3800, Australia; 8FAU Marine Science Laboratory, Department of Biological Sciences, Florida Atlantic University, Boca Raton, FL 33431, USA

**Keywords:** juvenile dispersal, green sea turtle, head-started releases, generalized additive mixed models, satellite tracking

## Abstract

**Simple Summary:**

Tracking juvenile animals is often difficult due to their small body sizes. We put miniature satellite tags on 40 captive-raised (“head-started”) green sea turtles (*Chelonia mydas*) and released them during two different times (January and July) off the Cayman Islands to determine whether there were differences in their movements. We used a statistical model and plots of turtle swimming directions versus ocean currents to find out what influenced their swimming direction and speed. We found that turtles released in January had very different movement patterns from those in July. The January turtles all moved in random directions from the islands around the Caribbean, whereas the July turtles all moved north of the islands. Additionally, we found that both the January and the July turtles swam harder and with more direction the closer they got to coastal areas. Turtles also appeared to mostly swim in the opposite direction to ocean currents to keep their desired path. These results show that captive-raised juvenile green turtles can likely move through deeper ocean waters after being released to coastal habitats where resources (such as food and shelter) are available. These results are important for future releases of captive-born turtles and for the conservation of wild Caymanian green turtles.

**Abstract:**

Juveniles of marine species, such as sea turtles, are often understudied in movement ecology. To determine dispersal patterns and release effects, we released 40 satellite-tagged juvenile head-started green turtles (*Chelonia mydas*, 1–4 years) from two separate locations (January and July 2023) off the coast of the Cayman Islands. A statistical model and vector plots were used to determine drivers of turtle directional swimming persistence and the role of ocean current direction. More than half (*N* = 22) effectively dispersed in 6–22 days from the islands to surrounding areas. The January turtles radiated out (185–1138 km) in distinct directions in contrast to the northward dispersal of the July turtles (27–396 km). Statistical results and vector plots supported that daily swimming persistence increased towards the end of tracks and near coastal regions, with turtles largely swimming in opposition to ocean currents. These results demonstrate that captive-reared juvenile greens have the ability to successfully navigate towards key coastal developmental habitats. Differences in dispersal (January vs. July) further support the importance of release timing and location. Our results inform conservation of the recovering Caymanian green turtles and we advise on how our methods can be improved and modified for future sea turtle and juvenile movement ecology studies.

## 1. Introduction

An ever-present and challenging goal in movement ecology is to describe the “lifetime track” or the entirety of movement phases for an individual throughout its entire lifetime [1]. Given that tag technology with the ability to report animal spatial–temporal positions is generally limited in duration to time periods less than a year, understanding the lifetime movements of any one individual has been all but impossible, although advances are being made for some taxa (e.g., white storks [2,3]). Instead, understanding ontogenetic movements, behavior, and habitat use has often relied on independently tracking individuals within different age classes and using quantitative methods combined with biological knowledge to infer connections between life stages. Additionally, active biotelemetry studies on more than one life stage have often been biased towards larger and adult animals, due to challenges with tagging neonates and juveniles, such as higher mortality rates, large tag-to-body-size ratios, and, for animals that disperse long distances, accessibility to intermediate age classes [4,5]. Of those studies that have successfully tracked these younger and smaller age classes (e.g., recently, neonate leatherback turtles [6]; juvenile green turtles [7]; juvenile bearded seals [8]; juvenile Scopoli shearwaters [9]; juvenile cuckoos [10]; juvenile king penguins [11]; juvenile silky sharks [12]; neonate caribou [13]), the challenges and expense of long-lasting tags have often resulted in smaller sample sizes, shorter tracking durations, and/or only one age class tracked at a time.

The complexity of life cycles for many migratory marine taxa and their presence within a dynamic, vast, and fluid medium has resulted in a paucity of understanding of juvenile movements at sea. Of particular interest is the sea turtle, a long-lived taxa where most species demonstrate ontogenetic separation in foraging habitats [14,15], as well as inter-specific and even population level differences in migration strategies [16]. They have a cryptic developmental life stage (previously termed “The Lost Years” [17]), where neonates and smaller juveniles spend years in pelagic foraging habitats far from both nearshore reproductive and foraging grounds. Although most sea turtle movement studies have focused on adult post-nesting females [5,16], a growing body of the literature has targeted understanding the “when” and “where” of movement [1] of this pelagic developmental period, including developmental habitat shifts that occur as larger juveniles.

Chelonian sea turtles (green, hawksbill, loggerhead, flatback, and Kemp’s and olive ridley) have been shown through a combination of mark–recapture, stable isotope, and tracking studies to transition from offshore pelagic developmental habitats (often in association with patches of drifting *Sargassum* algae) to nearshore benthic developmental habitats as larger juveniles [15,17,18,19]. Studies of the green sea turtle’s (*Chelonia mydas*) movement ecology have shown that, during their developmental pelagic stage, juveniles do not act as passive drifting particles but instead they actively orient and swim against ocean currents to remain in or to encounter favorable foraging habitats [20,21,22]. Studies on the navigational abilities and large-scale habitat use of juvenile sea turtles have mostly been limited to simulated individual based models (IBMs) and/or laboratory studies [23,24,25,26]. The progression and miniaturization of long-lasting satellite tags (now on the order of 10 g weight or less and capable of recording tracks over 100 days in duration) has recently allowed for empirical measurements of juvenile sea turtle movement [21,27,28,29,30]. However, the simultaneous tracking of multiple age classes is rare and, for many species and populations of sea turtles, the physical drivers and the role of ontogeny in dispersal is mostly empirically undescribed. Further work is needed to determine the relationship of juveniles’ directed movements with the physical environment (e.g., ocean currents, variable sea surface temperatures, and floating *Sargassum* algae communities) and to determine whether there are ontogenetic differences in these movements.

The Cayman Islands once hosted one of the world’s largest green turtle nesting colonies, but human over-consumption and exploitation drove the population to near local extinction by the late 1800s [31]. Recent research, however, has shown that previous mass releases (over 30,000 individuals, Bell et al., 2005) of neonate and juvenile captive-bred green turtles from the Cayman Turtle Conservation and Education Centre (CTCEC)—a breeding and captive center for green turtles of all age classes—most likely allowed for the successful reestablishment of a green turtle nesting population in the Cayman Islands [32]. Previous mark–recapture studies of these mass releases found that the turtles dispersed throughout the Caribbean before some returned to nest [33,34]. Although there are currently no satellite tracking studies of neonates and juveniles from the CTCEC or the local wild population, limited satellite tracking of adult wild Cayman green turtles has shown that they utilize foraging grounds off of Central America [35].

To investigate ontogeny in green turtle movement ecology and dispersal, we satellite-tagged and released captive-raised green sea turtles in multiple juvenile age classes from the Cayman Islands. Having been in captivity during the period of time that they would have been in their pelagic developmental stage, this experiment gave the unique opportunity to test the ability of captive reared or “head-started” juveniles to disperse effectively away from the release site and into a novel ocean environment. Turtles were released from two different locations and time periods, giving an additional opportunity to test for release effects. A generalized additive mixed model (GAMM) of turtle persistence swimming velocity (or the tendency of turtles to swim persistently in a given direction, km/day) was used with a mixture of abiotic and biotic covariates to determine whether there were (a) ontogeny in their directional movements, (b) differences in directional movements between release site locations and timing, and (c) relationships between turtle directional swimming and key environmental variables. Our study provides a novel contribution to the growing body of knowledge on movement ecology for juvenile life stages. To our knowledge, it is the first large tracking study performed on multiple age classes of juvenile green turtles in the Caribbean. Additionally, our work has significant implications for the conservation of the Cayman Island green turtle nesting population and of other imperiled sea turtle populations, with results informing the design of other ex situ conservation initiatives.

## 2. Methods

### 2.1. Satellite Tag Deployment and Turtle Releases

Juvenile green sea turtles were obtained from the Cayman Turtle Conservation and Education Centre (CTCEC) (Grand Cayman, Cayman Islands, Appendix A). The turtles were hatched and raised in a captive setting, reared in various sized concrete tanks (up to 150 m^2^, roughly I meter depth, supplied with direct unfiltered seawater), fed a nutritionally complete, floating, and modified extruded fish diet, and kept in groups of like size and age until they had reached the desired age class for release (further details on captive rearing can be found in the “Commercial Reptile Farming” section of *Mader’s Reptile Medicine and Surgery*, [36]). All turtles were quarantined prior to release and were checked to ensure that they each fulfilled requisite health and quarantine requirements (according to CTCEC quarantine and health screening protocol, available upon request from the CTCEC) for release.

Forty turtles across three different age classes (1–2 years, 2–3 years, and 3–4 years) were acquired for release in January and July of 2022 based upon availability (Appendix A). All turtles were designated to be in good health and body condition. All handling and procedures were approved by the University of Maryland Center for Environmental Science’s Institutional Animal Care and Use Committee (IACUC; research protocol No. S-CBL-2021-03). Turtles were either tagged with miniature (~1 g) solar (*N* = 15 in January, *N* = 10 in July) or battery-powered Lotek satellite transmitter tags (*N* = 15); the use of these different tags allowed for the comparison of tag performance between the two tag types but was also due to tag availability. The majority of tags were attached to the turtles’ carapaces using epoxy adhesive (*N* = 25), but some (*N* = 15, 1–2 years old) had 3M 5200 Fast Cure™ (FC) marine adhesive instead [22] (Appendix A). All tags (*N* = 40) reported positions and error information through the ARGOS satellite system (argos-system.org). Differences in tag performance (e.g., recorded tracking durations or daily positions reported) with tag and adhesive types were examined using appropriate statistical tests, with an analysis of variance (ANOVA) test to compare mean tracking duration between tag types (January solar tags, July solar tags, and January battery-powered tags), a repeated measures ANOVA to compare the mean number of positions reported per day between tag types (January solar tags, July solar tags, and January battery-powered tags), and two sample Student *t*-tests to compare mean tracking durations and mean number of positions reported per day for the adhesive types (Fast Cure™ vs. epoxy).

Based upon the availability of the turtles, individuals in different age classes were tagged and released during the two separate time periods (January and July 2022). Thirty turtles across three age classes (*N* = 10, 1–2 years, 35–40 cm CCL; *N* = 10, 2–3 years, 44–62 cm CCL; *N* = 10, 3–4 years, 54–62 cm CCL; Appendix A) were tagged and released on 22 January 2022 and ten additional turtles in the youngest age class (1–2 years, 36–40 cm CCL; Appendix A) were tagged and released on 20 July 2022. All age classes were released by boat off the coast of Grand Cayman, allowing them to be “jump started” to optimize release success and survival by avoiding nearshore predation. To compare impacts of releasing turtles in different locations, the January turtles were released ~ 10 km north of Rum Point, Grand Cayman, whereas the July turtles were released ~10 km south of Spotts Beach, Grand Cayman (Appendix A). Surface drifters (January, *N* = 1, Pacific Gyre REEF Drifter; July, *N* = 1, Microstar GPS Drifter) were released simultaneously with the turtles and reported regular (every 10 min) locations for use in understanding smaller scale surface current trajectories upon release and in the study region. Turtles were categorized as being either “dispersive” if their recorded tracks left the vicinity (~10 km) of Grand Cayman island or “residential” if their tracks remained in the vicinity of Grand Cayman island after release.

All data processing and analyses were performed in R version 4.2.0 [37].

### 2.2. Environmental Data Annotation

To match the resolution of environmental data products and to filter out erroneous positions, tracks of “dispersive” individuals (those that left the vicinity of the island, *N* = 22) had any relocations within a 10 km buffer of the island removed. All remaining locations were then further filtered and regularized through the “foieGras” R package [38], using the “fit_ssm” function. This function allows users to simultaneously filter and predict ARGOS locations at regular time intervals by fitting a simple random walk as a continuous-time (state space) process model [39]. We used error information from the tags (ARGOS error ellipse variables) and a 24-h time step (to match the temporal resolution of environmental data and to account for large gaps in positions) to predict daily positions for each turtle, with the resulting trajectories compared with raw locations to assess the potential for over-interpolation (Appendix A). Regularized trajectories for each individual turtle were then corrected for the influence of ocean currents using the methods of Gaspar et al. [40]. All current corrections were provided by the Copernicus Marine Service (https://marine.copernicus.eu/ accessed on 2 February 2022). Mean ground velocities were derived from daily observed positions; estimated surface current velocities derived from ocean current data were subtracted from these ground velocities to produce daily mean swimming speeds and current-corrected trajectories [40]. Ocean surface current data were obtained from the PSY4V3 product and the Operational Mercator global ocean analysis and forecast system (product DOI: https://doi.org/10.48670/moi-00016 accessed on 2 February 2022), which produces daily mean estimates of sea surface temperature and surface current velocity at a 1/12th degree (~9.2 km) resolution.

Environmental data for *Sargassum* density and sea surface temperature were additionally obtained for each daily turtle location. Mean daily sea surface temperature (SST) data were derived from the PSY4V3 ocean product. *Sargassum* density data were sourced from a processed product of *Sargassum* biomass density for the region (University of South Florida [41]), where weekly mean *Sargassum* biomass densities (g/m^2^) have been estimated at a 0.01 degree (~1.1 km) resolution from MODIS alternate floating algae index (AFAI) data and processed for errors from clouds and cloud shadows, sun glint, and gaps in satellite coverage [41]. Underlying *Sargassum* density values were extracted within a 1 km radius of daily turtle locations.

### 2.3. Graphical Assessments of Turtle Trajectories

To compare the initial magnetic headings (degrees from magnetic north) of turtles upon release between the two tracking periods (January and July), circular histograms were used on the first two recorded locations for each turtle with viable locations (*N* = 38; January: mean of 25 h ± 53 SD, July: mean of 18 h ± 10 SD). Initial turtle headings were compared with the initial direction of surface drifters to assess how turtles aligned with surface currents upon release.

Circular histograms were also used on regularized daily locations of all dispersive turtles (*N* = 22) to compare daily turtle headings with mean daily ocean surface current directions between tracking periods (January and July). Statistical differences between mean turtle headings (January vs. July) and mean ocean current directions were further examined using a Watson’s U^2^ two sample test with the “circular” R package [42].

Maps of individual trajectories and surface current fields were created for each tracking period (January and July) to compare dispersal patterns with respect to ocean currents and release site and timing. Daily mean ocean surface currents were derived from the PSY4V3 product and averaged over the maximum tracking duration for each tracking period (January: 28 days, July: 10 days) to provide background visualizations of mean current velocities. Surface current velocity fields were provided by the Copernicus Marine Service (https://marine.copernicus.eu/ accessed on 2 February 2022).

Plots of regularized trajectories for turtles that dispersed from the island (*N* = 22) were additionally visualized for each individual, with daily ocean surface current and turtle swimming vectors to determine where along each track turtles either swam in opposition to or in a similar direction and velocity to ocean surface currents.

### 2.4. Generalized Additive Mixed Model

A generalized additive mixed model (GAMM) was used for the regularized daily data for all dispersive turtles (*N* = 22), with a mixture of abiotic and biotic covariates to determine drivers of daily swimming persistence in a given direction. Daily swimming persistence velocities were determined for each turtle using daily turning angles (in radians, the angle between two successive swimming speed vectors) and step lengths (in km day^−1^, straight line displacement between two successive locations) for current-corrected trajectories. Daily turning angles and step lengths were derived using the “amt” R package [43]. The daily persistence velocity, or the tendency and magnitude of a given movement to persist in a given direction on a daily scale, was then determined by multiplying the magnitude of each daily step length by the cosine of the corresponding daily turning angle [44].

Potential drivers of turtle daily persistence velocity included in the model selection process were (on a daily resolution): mean ocean surface current velocity, sea surface temperature, *Sargassum* biomass density, turtle size (body weight), and the tracking period (January or July). These were included in the model selection process as fixed effects. To account for individual replication with the variable for turtle size, individual turtles were included as a random effect. To account for spatial autocorrelation, a smoother interaction of latitude and longitude was included. To account for temporal autocorrelation, a correlation autoregressive moving average (ARMA) structure was included in candidate models, with the initial values for *p* and *q* determined using autocorrelation function (ACF) and partial autocorrelation function (PACF) plots.

Model selection to determine the covariates most influential on the daily persistence velocity of turtles was performed using a standard approach for mixed effect models, whereby an initial model with all possible covariates and structures was fit using maximum likelihood estimation (MLE) and compared with successive models with varying combinations of fixed effect covariates. Backward elimination using the Akaike information criterion (AIC) was performed for these candidate models to select an appropriate fixed-effect structure. This approach was also used to determine whether an ARMA structure was appropriate and, if so, to select the best structure for *p* and *q* based on the fits from a restricted maximum likelihood estimation (REML). The final selected model was re-fit with the “mgcv” R package [45] and model assumptions were checked using a plot of the standardized residuals versus the fitted values. The final model results were visualized by plotting GAMM predicted values for daily persistence velocity against fitted values for selected covariates. Predicted persistence velocity values were visualized as a function of the non-linear interaction of latitude and longitude, using a partial effects contour plot from the “itsadug” R package [46].

## 3. Results

### 3.1. Tag Summaries

Of the 30 turtles released in January, two (Appendix A) reported only one location, each of which were outside the boundaries of Grand Cayman island; these turtles were dropped from further analysis. Of the remaining turtles released in January with viable positions (*N* = 28) upon release, all moved in a primarily southerly direction back towards Grand Cayman island, generally matching the direction of the surface drifter (Figure 1A) and returning to the vicinity of the island within 24 h of release. Twelve turtles then dispersed from the island in a radial fashion and were labeled “dispersive” (Table 1, Figure 1C), whereas the remaining 16 turtles remained within a 10 km buffer (Appendix A) of the island for the duration of their tags and were labeled “residential”. The January surface drifter washed up on Grand Cayman and was re-released on 26 January 2022.

Of the 10 turtles released in July, all immediately dispersed away from the island (were “dispersive”) and had viable positions, with initial movements reflecting the direction of the surface drifter northwest of the island (Table 1; Figure 1A,D).

Of the 22 turtles that dispersed from the island in both January and July (“dispersive” individuals, Figure 1C,D), there was an uneven distribution in the number of individuals represented from each age class (*N* = 5, 1–2 year olds, *N* = 4, 2–3 year olds, and *N* = 3, 3–4 year olds in January; *N* = 10, 1–2 year olds in July) (Table 1). Of the remaining 16 turtles that stayed within the vicinity (10 km) of Grand Cayman for the duration of their tags (“residential”), there was also an uneven distribution in the age classes (*N* = 5, 1–2 year olds, *N* = 4, 2–3 year olds, and *N* = 7, 3–4 year olds) (Appendix A). Dispersive turtles reported an average of 12.6 positions per day (range: 2.6–17.4 mean positions per day), whereas turtles remaining near the island (“residential”) reported an average of 4.2 positions per day (range: 1.0–9.1 mean positions per day) (Appendix A).

Results of an analysis of variance model (ANOVA) showed no significant differences (F-value: 0.48, *p*-value: 0.62) in the tracking duration between the tag types (January solar tags, July solar tags, and January battery tags) (Appendix A). A repeated measures ANOVA did find, however, that the solar tags reported a significantly higher average number (6.4 ± 4.7 SD for January release, 12.8 ± 5.1 SD for July release) of positions per day compared with the battery-powered tags (2.5 ± 1.3 SD, only in January) (F-value: 54, *p*-value: 3.8 × 10^−11^) (Appendix A). For the different adhesive types tested with the tags on an equal number of 1–2 years olds released in January, although turtles with Fast Cure™ adhesives had a longer mean tracking duration of 14.4 days (± 7.4 SD) compared with 8.7 days (± 1.8 SD) for those with epoxy adhesives (Appendix A), a two sample Student’s *t*-test did not support a significant difference (t-value: −1.5, *p*-value: 0.18). Fast Cure™ attachment types for 1–2 year olds released in January had a similar average number of positions reported per day (6.3 ± 5.1 SD) compared with the epoxy tags (6.4 ± 5.5 SD) (Appendix A), with a two sample Student’s *t*-test finding no significant difference (t-value: 0.031, *p*-value: 0.98).

### 3.2. Graphical Assessments of Turtle Trajectories

Circular histograms and statistical tests of daily magnetic headings for turtles that dispersed from the island (*N* = 22, “dispersive” turtles) showed significant differences in mean headings between turtles released in January vs. July (Watson’s U^2^ two sample test, t-value: 0.620, *p*-value: <0.001). A large proportion (42% or 5/12 turtles) of turtles released in January had a mean southwest heading throughout their dispersal, whereas a large proportion (50% or 5/10 turtles) of those released in July had a mean northeast heading (January: 226 deg ± 101 SD, July: 20 deg ± 94 SD; Figure 1B). Significant differences were further seen between the mean headings of turtles and the mean direction of ocean currents for each tracking period (Watson’s U^2^ two sample test, January t-value: 0.703, July t-value: 1.101, January and July *p*-values: <0.001) (Figure 1B). Mean current directions experienced by turtles along their tracks were also significantly different between tracking periods (Watson’s U^2^ two sample test, t-value: 0.402, *p*-value: <0.001), with currents for the July turtles having a predominantly northwest direction in comparison to currents reported in January (January currents: 288 deg ± 76 SD, July currents: 331 deg ± 75 SD; Figure 1B).

Maps of mean current velocity fields showed that the trajectories of turtles released in January often crossed over regions with high current velocities (>0.7 m·s^−1^) and radiated out in distinct directions from the island (Figure 1C). In July, the turtles overall dispersed into similar areas north of the island, where current velocities were slower (<0.5 m·s^−1^) (Figure 1D).

Plots of daily ocean current and turtle swimming vectors over regularized trajectories for both January and July showed that the turtles largely swam in opposition to the currents, with similar magnitudes as current vectors, to maintain their trajectories (Figure 2A,B). However, in some cases (e.g., IDs 203411, 212847, 229671, and 229678), the turtles swam in the same direction as the currents (Figure 2A,B).

### 3.3. Generalized Additive Mixed Model

The model selection process resulted in a final GAMM of turtle daily persistence velocities as a function of the fixed effect of the tracking period (January and July) and the smoothed interaction of latitude and longitude, with an ARMA structure of *p* = 4 and *q* = 4 to account for temporal autocorrelation (Table 2).

Predicted values for daily persistence velocity had similar means but distinct standard deviations between tracking periods (January: 2.0 km day^−1^ ± 19 SD, July: 1.8 km day^−1^ ± 9.0 SD). A plot of these predicted values by individual and colored by tracking period further showed a larger variation between individuals and a higher median for turtles released in January (January: 4.4 km day^−1^, July: −0.05 km day^−1^, Figure 3A). An additional plot of predicted persistence values over time showed that the majority of individuals in both tracking periods had persistence velocities closer to zero at the beginning of their tracks and an increase in persistence velocity over time (Figure 3B). In January, most turtles (9/12) persisted in their westerly velocity component (positive persistence velocity values) over time, whereas in July turtles persisted in their easterly velocity component (negative persistence velocity values) or had persistence velocities approach zero over time (Figure 3B). Contour plots of the partial effects for the interaction of latitude and longitude showed a strong longitudinal gradient in predicted persistence velocities over the study area (Figure 3C), with smaller values in early portions of the tracks and near Grand Cayman island and Cuban shelf areas and larger values towards the end of the tracks, especially for those seemingly headed to coastal and nearshore areas of Cuba and Central American countries such as Honduras, Costa Rica, and Panama. (Figure 3C).

## 4. Discussion

This study, to our knowledge, represents the first large scale satellite tagging effort of juvenile green turtles in the Caribbean. Despite limitations in tag durations, our results support that head-started juvenile green turtles can successfully disperse from release locations. Many of the individuals in our study demonstrated an ontogenetic niche shift, either swimming with persistence through pelagic waters towards nearshore areas around the Caribbean (“dispersive” turtles) or remaining nearshore of the Cayman Islands for the duration of their tracks (“residential” turtles). This supports that, with longer tracking durations, turtles would likely have been observed to recruit long-term to a variety of nearshore areas known to serve as developmental habitats around the Caribbean and the Cayman Islands. Although we tagged multiple age classes of juvenile turtles, our model did not detect significant differences in dispersal behavior between age classes. Behavioral differences amongst turtles largely coincided with where and when the turtles were released. This highlights the significance of release sites and timings for captive-bred green turtles. Our results have important implications for captive-reared studies on turtles, as our observed dispersal differences may have long-term implications on turtle survival if they result in turtles recruiting to habitats where threats (e.g., bycatch or illegal take by fisheries) are more acute. Our study provides an approach to understanding the movement ecology and behavior of juvenile animals that could be modified for other taxa with poorly understood migratory juvenile life stages.

### 4.1. Effects of Release Location and Timing on Dispersal

Remarkable differences were seen in the dispersal trajectories of turtles released at different sites and times of year, potentially resulting from distinct current structures at the release locations. Simultaneous releases of surface drifters with the turtles revealed that turtles in both time periods generally followed the direction of surface currents upon release (Figure 1A), with turtles in January moving south of their release location back towards the island and those in July moving north-west of their release location towards waters north of the island. These early differences between January and July releases are evidence of how favorable conditions (e.g., surface currents) can aid the dispersal of turtles away from the release point [47]. All turtles released in January (*N* = 28, after two individuals with unviable positions were removed) appeared to return after release to the vicinity (within 10 km) of the island, with most (*N* = 16) remaining in this area for the duration of their tag’s lifetime. These turtles may have eventually left the vicinity of Grand Cayman island, but obtaining records of such dispersal was not possible due to many potential behavioral factors, such as turtles scraping or removing tags on hard surfaces around the island [48] or early tag failures (tag duration ranged 3.0–32.5 days). Another possibility, although additionally impossible to determine due to the shortness in tag durations, is that some or all of these turtles remaining near the island took up residential behavior and recruited to local habitats. All of the “residential” turtles that remained near the island were of body sizes that have been recorded in another study (mean: 55.7 cm CCL, range: 32.8–80.7 cm CCL [49]) and inhabited a local southern lagoon, where diverse behavior (e.g., moving in and out of the lagoon and diurnal seagrass and coral reef use) was observed. Differences seen between the average number of positions reported per day for “dispersive” versus these “residential” turtles in our study, with far more positions reported per day on average for “dispersive” turtles (mean: 12.6 positions per day) compared with “residential” turtles (mean: 4.2 positions per day), further supports that the 16 turtles remaining near the island for the duration of their tags may have been performing residential behaviors, such as less frequent diving and more benthic resting or foraging behavior. Interestingly, there was a higher number of individuals within the larger age class of our study (3–4 year olds, 54–62 cm CCL, Appendix A) of these “residential” turtles (*N* = 7, 3–4 year olds) versus the “dispersive” turtles (*N* = 3, 3–4 year olds). These older and larger juveniles (close to the mean size of 55.7 cm CCL observed by Blumenthal et al., [49]) may have been more likely to recruit locally due to being closer in size and age to when the life stage shift between pelagic to neritic developmental habitat occurs [15,18,34]. Overall, although we were limited due to tag durations in being able to conclude long-term behaviors and residency of our released turtles, these findings emphasize the importance of currents when choosing locations for the release of marine and captive-reared species. Given that the turtles in both tracking periods, regardless whether they remained near the island or not, initially responded strongly to the direction of the currents, conservation programs aiming to optimize tracking durations of released captive-reared individuals could potentially benefit from long-term sea turtle movement modeling (e.g., STAMM,25) or simulations set at different times of year to determine the best site and time of year for release [47,50].

The effect of the differences in release location and timing, whether it be positively or negatively beneficial for turtle long-term survival and migration, is difficult to discern due to the short duration of the tracks (mean 15.2 days in January and 10.6 days in July). Only 12 of the turtles released in January were observed to disperse away from the island (after a period of ~4.5–19.5 days), whereas all 10 turtles released in July dispersed away immediately from both their release location and the island. Often a goal in captive-bred releases is for individuals to effectively disperse away from the release location into novel environments [51]. However, although there was likely a risk of human poaching [32] and potentially competitive interactions for resources with other older resident turtles, the Cayman Islands do support an active local wild population of green turtles [32,33]. Resources are available for those that stayed near the vicinity of the island and it is possible that the 16 “residential” turtles released in January recruited to local Caymanian foraging and resting habitats [49].

The effects of differences in release location and timing could be hypothesized with respect to their similarity to “soft” versus “hard” releases for other captive animals. These are common methods in terrestrial studies, where “soft” releases allow animals to acclimate to their surroundings and can have higher success in contrast to “hard” releases, which have no acclimatization period [52]. The effects of “hard” vs. “soft” releases for captive-raised aquatic and marine animals are not as well studied [52], but are probably best known for marine mammals [53,54,55], where “sea pens” or enclosed areas in their natural habitat can be used to acclimate the animals to the ocean environment before a full release, potentially minimizing any harmful effects of sudden release (e.g., startling, confusion and an inability to find food, [52]). For our study, turtles released in January may have had more of a “soft” release, with ocean currents leading them back to the island and giving them time to familiarize themselves with their environment and even forage before either remaining as residents in local waters of the Cayman Islands or leaving at a chosen time to migrate to other developmental habitats in surrounding geographic areas (e.g., Central America or Cuba). The distinct dispersal trajectories around the Caribbean for turtles dispersing from the island in January (Figure 1C) may have beneficial impacts on their long-term survival by “spreading out” recruitment to coastal and shelf areas in the Caribbean, reducing risks of overcrowding and competition for resources at these important sites and potentially minimizing any one individual’s risk of harmful human interactions (e.g., bycatch or poaching). In contrast, turtles released in July with more of a “hard” release immediately seemed to follow currents away from the island before dispersing into new areas north of the islands (Figure 1D). While this certainly follows some criteria of a successful release by evidencing immediate dispersal away from the release site [51], only 2/10 of the individuals released in July seemed to have persistent movements toward productive coastal recruitment areas of the Cuban shelf (Figure 3C). Instead, most seemed to be caught up in an area characterized by a clockwise circulation called the “Cuban Vortex”, an expansion of the Loop Current [56]. This current structure may have influenced their movements away from these recruitment areas and, indeed, although ocean surface current fields had lower velocities in this area than those experienced in January, turtles swimming within the Cuban Vortex area in July primarily swam in opposition to current directions (Figure 2B) and had less directed movements than those in January (Figure 3A–C).

### 4.2. Individual and Current Influences on Dispersal

In addition to differences in movement patterns with release period (January or July), our results showed that turtles had individual variability in their dispersal. Multiple turtles in January (IDs 203084, 203408, and 203409) crossed over areas with high current speeds (>1 m·s^−1^, Figure 1C), swimming in opposition to the currents with oriented swimming to maintain their southwest trajectories towards Honduras and Nicaragua (Figure 2A; 21, 28, 40). Others (IDs 212867, 214069, and 214085) similarly swam in opposition to the direction of the currents (Figure 2A) to move across higher velocity areas in the Cuban Vortex towards the Cuban continental shelf. In contrast, a few individuals seemed to swim into the direction of the currents (IDs 212847, 229671, and 229678, Figure 2A,B), perhaps allowing the current advection to “boost” their dispersal and save energy reserves [28,40].

Although turtle swimming appeared to be significantly responsive to ocean current direction in both tracking periods (Figure 2A,B), our statistical model found that turtles demonstrated a directed drive to their movements independent of current speeds, sea surface temperature, *Sargassum* algae density, or body size. Instead, turtles significantly varied their movement persistence with longitude (Figure 3C) and time along their track (Figure 3B), with turtles in January on average increasing their movement persistence towards the end of their tracks and as they approached areas closer to coastal and shelf recruitment areas. Although no model fully captures the system complexity (there is a chance that temporal and spatial autocorrelation were not fully accounted for), our final model (Table 2) did include both spatial and temporal autocorrelation terms to best minimize bias from either of these. An increased movement persistence as turtles neared land masses and coastal foraging areas around the Caribbean might indicate an increased navigational sense, where a combination of possible cues (e.g., olfactory, auditory, chemical, visual, or waves) may help direct them further nearshore and in opposition to current drift [57,58] (Hays et al., 2003, Lohmann et al., 2008). For turtles in July, four individuals were recorded to essentially return “home” (back to the vicinity of the Cayman Islands) by the end of their tracks (IDs 229669, 229672, 229676, and 229677), broaching the possibility of homing or imprinting on the rearing location for captive-raised and/or juvenile turtles [59]; indeed, all four of these individuals appeared to have swimming vectors directed towards the islands towards the end of their tracks (Figure 2B). These four individuals were within the youngest age class (1–2 years); however, as none of the 1–2 year old individuals released in January demonstrated this possible homing behavior, it may be more of a signal of the release strategy and current structures around the island in July, rather than the size or age of the turtles.

We were not able to test for relationships of turtle movements and behaviors with the magnetic field, but we cannot rule out this possibility, as previous laboratory studies have found evidence that turtles may use locational markers of the geomagnetic field (inclination and intensity) to imprint on and return to natal and foraging sites [60,61,62]. Two of the “homeward bound” individuals in July (IDs 229676 and 229677) seemed to demonstrate a complete reversal in their trajectory directions after ~6–7 days (Figure 1D and Figure 2B), only traveling a half of degree of latitude or less before reversing direction; this could be evidence of a “confusion” effect from the potential “hard” release in July and/or the turtles may have found a resource (e.g., *Sargassum* mats) and used active swimming to remain in the patch and prevent drift from the currents [22,28,40].

### 4.3. Evidence of Long-Term Nearshore Recruitment

Although our statistical model did not detect a significant response between turtle swimming persistence velocity and *Sargassum* density, this may have been an artifact of both the coarse resolution of the movement data used in the model (daily) and individual variability in movement persistence (Figure 3B). Opportunistic foraging on the movement path can be a fine-scale behavior that is easily masked by a daily interpolation of locations [63], such as with the data for our turtles. Additionally, although the pelagic developmental stage for green turtles is thought to last on average 3 years (but with a range of 1–7 years [18,64]), it is common for animals raised in captivity to have larger body sizes than their wild counterparts [65,66,67]. The turtles in our study (January: 1–2, 2–3, and 3–4 years, CCL range: 35–65 cm; July: 1–2 years, CCL range: 36–40 cm) were all within the size range for juvenile green turtles in the transitional recruitment phase that are shifting from pelagic to nearshore benthic developmental habitats [15,18,34]. Despite the resolution of our data potentially restricting our ability to detect a relationship between *Sargassum* density and turtle swimming persistence, floating patches of *Sargassum* algae are considered a key foraging resource for juvenile green turtles in the pelagic stage [19] and it is possible that juveniles foraged or associated with these patches during dispersal. Additionally, as previously discussed (see Section 4.1, Effects of Release Location and Timing on Dispersal), the majority of turtles released in January remained near the vicinity of Grand Cayman island for the duration of their tags (*N* = 16) and were all within the size range for other juvenile greens (mean: 55.7 cm CCL; range: 32.8–80.7 cm CCL [49]) that have been previously recorded to inhabit local nearshore lagoons, sea grass beds, and coral reefs [49]. Further research using tags with longer tracking durations and possibly combined with passive acoustic arrays or mark–recapture studies [49] is needed to determine whether head-started juvenile greens are recruiting to local foraging and resting habitats of the Cayman Islands.

It is likely that multiple individuals in our study eventually recruited to other nearshore developmental habitats around the Caribbean, with the majority (over 68% or 15 out of the 22 turtles) of dispersive turtles in our study (CCL range: 35–65 cm, Figure 1C,D) having trajectories towards other coastal habitats after leaving the vicinity of the Cayman Islands. Results from previous mark–recapture studies of juveniles and adults, combined with the results of small satellite tagging studies of adults, have shown that turtles sourcing from the Cayman Islands will migrate to areas off Cuba, Nicaragua, and Honduras, likely for high quality foraging and developmental habitats [33,34,35]. Turtles in our study may also have been opportunistically foraging along their tracks, especially turtles in July that had more of a “hard” release, where, unlike those in January that first returned to the island to potentially forage around its vicinity before dispersing, individuals instead took a minimum of 7 days (e.g., ID 229678, Figure 1D) to reach productive shelf areas of Cuba [34]. This is a time period that, if fasted completely, would likely result in drastically lowered metabolic rates that would hinder dispersal [68].

### 4.4. Implications for Head-Starting and Conservation Initiatives

The results of our study support that captive-reared (“head-started”) individuals have the ability to use directed movements and a variety of behaviors to navigate through an unknown environment to potential foraging and developmental recruitment areas. This finding is important both for this species and for this specific population of green sea turtle, a wild nesting population slowly recovering from decades of overhunting within and around the Cayman Islands [32]. These results are a demonstration of how similar ex situ head-starting studies could be useful for other more critically endangered species, such as the Eastern Pacific leatherback turtle [69,70]. Head-starting and release initiatives, where animals are born or hatched in captivity, raised to an age where their larger body sizes make natural mortality less likely, and released into the wild, have often been deemed unsuccessful ways of increasing sea turtle survivorship [71,72] and potentially disruptive to turtles’ natural navigational abilities, natal imprinting, and nesting site choices [59]. However, these criticisms may largely be due to a lack of monitoring and/or tracking efforts for most head-starting projects that can support evidence of long-term success [73]. Additionally, these studies may not account for the effects of oceanography (e.g., ocean currents) on dispersal. As our results show, release location and timing with respect to ocean current directions have potentially important implications for where released head-started turtles end up, with January individuals in our study following the direction of ocean currents back to their rearing location but July turtles following these ocean currents away from their rearing location. Our study addresses one of the key concerns about head-starting and captive-reared programs, i.e., that individuals will not be able to effectively navigate, forage, or recruit to important habitats due to being raised and/or kept in a synthetic environment [51]. Indeed, individuals in our study demonstrated the ability to disperse independently towards key habitat areas using a diversity of dispersal paths and behaviors. During January especially, and for those that dispersed from Grand Cayman island, this resulted in individual spatial separation in foraging and developmental habitats that may provide resilience to the population in the face of high anthropogenic mortality or inter/intra-specific competition for resources in these areas. Further, although most individuals released in January seemed to follow ocean currents to return and stay near the vicinity of Grand Cayman island for the duration of their tags, as previously discussed, there are known habitats capable of supporting the development of juvenile greens around the islands [49]. Their possible long-term residency may help support recruitment to the local nesting population of Caymanian green turtles, which has been estimated to have only 100–150 nesting females [32]. Longer-term monitoring, perhaps with the additional use of passive tags or mark–recapture efforts, is needed to determine whether individuals released in our study meet the criteria established for “successful” head-starting programs [51]. These include the ability to fully recruit to nearshore developmental habitats and mix with conspecifics, survival long term, and philopatry to their natal site. Recent research of past releases of captive-reared neonate, juvenile, and adult Caymanian green turtles support that they certainly have the capacity and potential to meet all of these criteria [32,33,34,35].

Longer tracking studies with even larger sample sizes and more age classes (e.g., neonates and adults) are needed to quantify spatial overlap and to potentially mitigate harmful interactions between fisheries and turtles in our study region within the Caribbean, both for turtles at sea and in nearshore foraging and inter-nesting habitats. Although the small and novel satellite tags we used in this study unfortunately did not have durations long enough to determine spatial use of nearshore areas, their duration was sufficient to predict the final destination of turtles that dispersed from the islands, which included nearshore and shelf areas of Cuba, Grand Cayman, and Central America (e.g., the Caribbean side of Honduras, Costa Rica, and Panama). All of these regions host intense fishing efforts [74], where green turtles are likely to be caught incidentally as bycatch and to be intentionally hunted (both legally and illegally, through poaching) for human consumption [32,33,34]. Although repopulation and mass head-starting efforts of the decimated local green turtle populations (such as by the Cayman Turtle Conservation and Education Centre or CTCEC on Grand Cayman) have been successful in starting to restore local and wild nesting populations [32,33], predictive management tools that combine near real-time movement and habitat use data with threat data (termed “dynamic management” [75,76]) could offer effective and complimentary approaches for mitigating bycatch and reducing illegal hunting both in oceanic habitats and within nearshore shelf areas for this population of green turtles.

## 5. Conclusions

To better understand and be able to quantify fine-scale behaviors, such as foraging, future studies should aim to reduce locational errors and maximize tracking durations by refining the novel miniature tags used in this study. Tags overall reported large amounts of error in positions (especially for the January turtles, Appendix A), with the battery-powered tags having a significantly lower number of positions reported compared with the solar-powered tags (Appendix A). Future studies should work with modified versions of these tags, prioritizing the solar-powered ones based on these results. However, despite these challenges, we were able to attain enough locations for each individual with our data to predict daily movements and overall trajectories (Appendix A). We acknowledge that the accuracy of these predictions was of course dependent on the data, which for some individuals with especially low quality or relatively long gaps may have resulted in over-interpolation or unrealistically straight portions of the track (e.g., for IDs 203,409 and 212867, Figure 1C and Appendix A). Despite these limitations, we believe these results provide valuable coarse-scale predictions of dispersal and behavior for juvenile green turtles in this region, representing the first empirical large sample size tracking study for this species in the Caribbean. This study will help to inform future investigations into the drivers of juvenile dispersal and swimming persistence for finer scales of movement.

The methods we used to determine ecologically significant movement-based behaviors could be useful for future movement ecology studies of this and other species. Future and longer tracking studies of this population, and on other migratory animals that move through a fluid environment (e.g., other aquatic or aerial species), that are able to attain finer-resolution data could extend our statistical model of a singular movement behavior to incorporate multiple movement behaviors using more rigorous and complex behavior parsing methods, such as discrete [77,78,79] or continuous time [39,80] state–space movement models, that can simultaneously account for hierarchical structures to the data, temporal autocorrelation, locational errors, and relationships with environmental variables. These methods could be taken further by using individual based models (IBMs) to model long-term dispersal and survival probabilities using both environmental variables (e.g., ocean currents or sea surface temperature) and habitat-directed movements. Models could be both parameterized by and compared with observations from empirical tracking studies [29,81]. Examples of IBMs such as this already exist for sea turtles (e.g., STAMM) and, although they have been successfully applied to other species (e.g., leatherbacks, [25,26]), their application to Caribbean green and head-started turtles could be useful in lieu of expensive tracking studies and limited availability of multiple age classes of turtles for releases.

Overall, this study represents, to our knowledge, the first large-sample size biotelemetry tracking effort of green sea turtles across multiple juvenile age classes in the Caribbean. Despite the limitations of our tagging data, these results support that juvenile green turtles demonstrate a diversity of dispersal patterns and have the potential to successfully recruit to productive nearshore areas similar to their wild counterparts. Our results also importantly inform ongoing conservation efforts and future tracking studies for sea turtles. We advise that future release studies should decide carefully where and when to release post-captive turtles. We share these findings to additionally provide an example of how similar methods may be used for biotelemetry-based studies of captive-reared (head-started) juveniles of other species, as these studies are still underrepresented in the fields of animal behavior and movement ecology.

## Figures and Tables

**Figure 1 animals-13-01218-f001:**
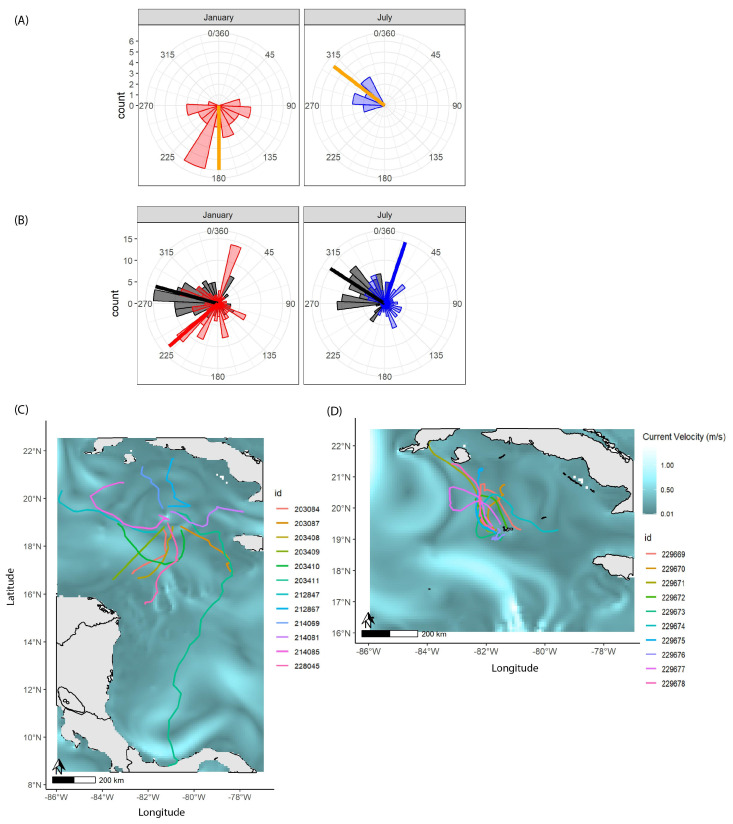
(**A**) Circular histograms of the initial heading (first two reported locations) of turtles released in January (*N* = 28, red) and turtles released in July (*N* = 10, blue), with the initial direction of the surface drifter shown (orange). (**B**) Circular histograms of overall mean daily surface current directions (black) versus January track (blue) headings and July track (red) headings, with thick colored lines showing circular means for each group. Trajectories for dispersive turtles in January (*N* = 28) (**C**) and July (*N* = 10) (**D**) plotted over ocean surface current fields averaged over the maximum duration of the tracks from each tracking period (January: 28 days, July: 10 days). Surface current velocity fields were provided by the Copernicus Marine Service (https://marine.copernicus.eu/ accessed on 2 February 2022).

**Figure 2 animals-13-01218-f002:**
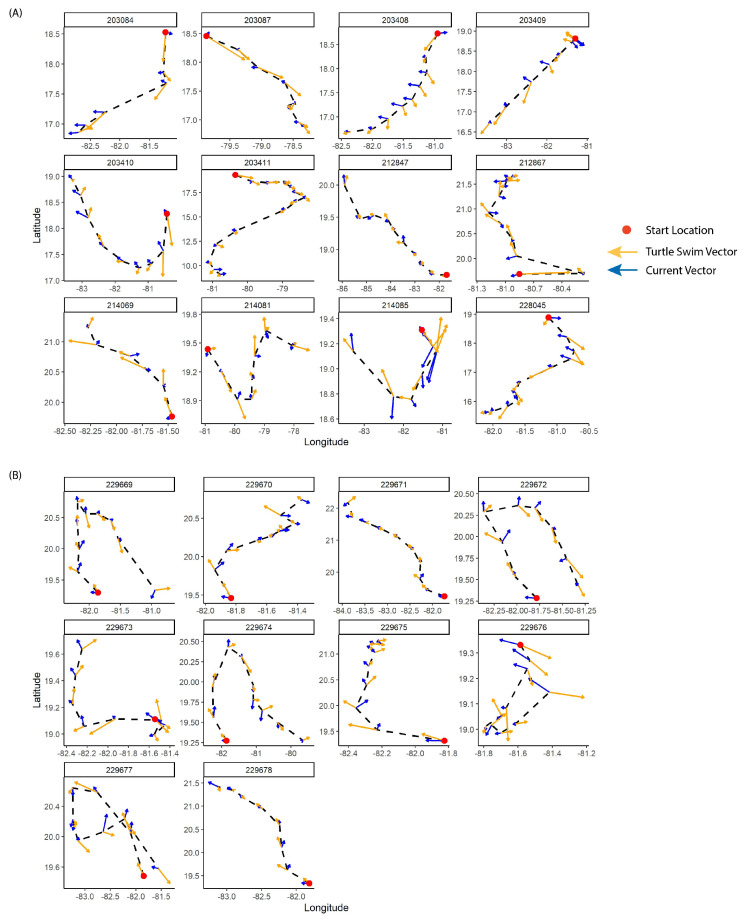
Plots of individual trajectories for turtles that dispersed from Grand Cayman island (*N* = 22), with daily turtle swimming (yellow) and ocean current (blue) vectors overlaid for (**A**) January and (**B**) July tracking periods. Red dots indicate start locations of trajectories.

**Figure 3 animals-13-01218-f003:**
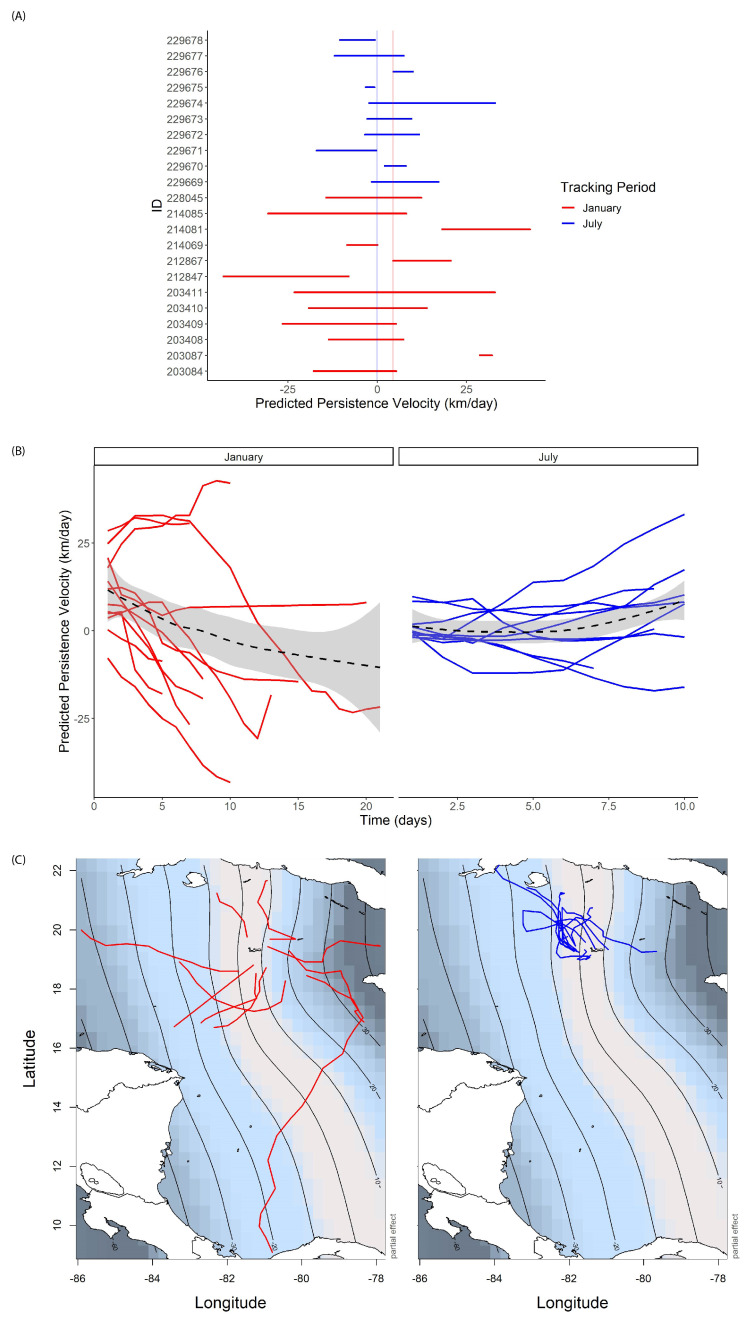
Predicted swimming persistence velocities from a fitted generalized additive mixed model (GAMM) for (**A**) individual turtles colored by tracking period (January, red; July, blue), with vertical bars showing median persistence velocity values for each tracking period; (**B**) individual turtles in January (red) and July (blue) over time, with generalized additive model smoothers (black line) and 95 percent confidence intervals (gray); (**C**) contour maps of the interaction of latitude and longitude with turtle trajectories overlaid for January (red) and July (blue).

**Table 1 animals-13-01218-t001:** Summary statistics for the daily locations of dispersive individuals (*N* = 22), with columns for: tracking period (January and July), age class (1–2, 2–3, and 3–4 yrs), track duration (days, after positions near island were removed), mean 24-h displacement (km), mean heading (0–360 degrees from true north), mean sea surface temperature (deg C), mean surface current velocity (km/day), mean swimming speed (km day^−1^), and total displacement (km; from start location to end of track).

ID	Tracking Period	Age Class	Track Duration days	Mean 24-Hr Displacement km	Mean Bearing 0–360 deg	Mean Sea Surface Temp deg C	Mean Current Velocity km day^−1^	Mean Swimming Speed km day^−1^	Total Displacement km
203084	Jan	1–2 Yrs	6	52 (±39)	178	27 (±0.20)	29 (±10)	53 (±22)	243
203408	Jan	1–2 Yrs	9	35 (±9)	216	27 (±0.16)	27 (±9)	38 (±17)	271
203410	Jan	1–2 Yrs	9	58 (±22)	275	27 (±0.17)	38 (±12)	54 (±38)	309
214069	Jan	1–2 Yrs	6	41 (±13)	278	27 (±0.36)	10 (±5)	42 (±10)	185
214085	Jan	1–2 Yrs	9	44 (±33)	226	27 (±0.36)	22 (±9)	57 (±29)	235
203087	Jan	2–3 Yrs	8	44 (±26)	145	27 (±0.23)	13 (±5)	48 (±42)	240
212847	Jan	2–3 Yrs	11	49 (±10)	264	27 (±0.33)	28 (±10)	27 (±12)	470
212867	Jan	2–3 Yrs	21	16 (±22)	80	26 (±0.37)	16 (±7)	28 (±23)	220
214081	Jan	2–3 Yrs	11	38 (±28)	146	27 (±0.38)	27 (±13)	46 (±32)	310
203409	Jan	3–4 Yrs	8	43 (±34)	223	27 (±0.19)	33 (±10)	58 (±23)	323
203411	Jan	3–4 Yrs	22	65 (±27)	172	27 (±0.24)	27 (±12)	60 (±22)	1138
228045	Jan	3–4 Yrs	16	29 (±30)	211	27 (±0.13)	23 (±14)	38 (±33)	382
229669	Jul	1–2 Yrs	10	37 (±12)	170	29 (±0.13)	18 (±9)	18 (±15)	96
229670	Jul	1–2 Yrs	10	18 (±14)	97	29 (±0.10)	10 (±5)	29 (±11)	151
229671	Jul	1–2 Yrs	10	41 (±12)	295	29 (±0.41)	26 (±7)	35 (±16)	396
229672	Jul	1–2 Yrs	9	31 (±14)	183	29 (±0.13)	17 (±7)	23 (±14)	52
229673	Jul	1–2 Yrs	9	22 (±13)	193	29 (±0.19)	20 (±4)	42 (±10)	91
229674	Jul	1–2 Yrs	10	43 (±12)	141	29 (±0.17)	16 (±5)	22 (±15)	233
229675	Jul	1–2 Yrs	10	27 (±18)	206	29 (±0.11)	15 (±4)	25 (±13)	213
229676	Jul	1–2 Yrs	10	11 (±5)	143	29 (±0.15)	19 (±8)	25 (±11)	27
229677	Jul	1–2 Yrs	10	47 (±23)	191	29 (±0.23)	22 (±9)	44 (±23)	32
229678	Jul	1–2 Yrs	7	38 (±13)	319	29 (±0.20)	21 (±6)	24 (±13)	270

**Table 2 animals-13-01218-t002:** Results of the final fitted generalized additive mixed model (GAMM) for turtle swimming persistence velocity (km day^−1^, Gaussian error, identity link function). Final model structures were selected using AIC comparison. The final model had a linear term for tracking period (July as the reference level to January) and a non-linear smoothing and interaction term for location (latitude, longitude). The model also included individual turtle tracks as random effects and accounted for temporal autocorrelation using a correlation autoregressive moving average (ARMA) structure with *p* = 4 and *q* = 4.

Response Variable				
	** *Fixed Effects* **	** *Estimate* **	** *Estimate Error* **	** *p-value* **
	*Intercept*	−0.7	1.9	0.70
Persistence Velocity	Tracking Period (*January*)	6.2	3.8	0.10
	** *Smoother Term* **	** *EDF* **	** *F-statistic* **	** *p-value* **
	Latitude, Longitude	6.3	17	<2 × 10^−16^

## Data Availability

Restrictions apply to the availability of these data. Data were obtained from Upwell Turtles and the Cayman Turtle Conservation and Education Centre (CTCEC) and are available (by request to the corresponding author and George Shillinger, upwell.org) with the permission of Upwell Turtles.

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
