# Peer review of "Satellite Tracking of Head-Started Juvenile Green Turtles (Chelonia mydas) Reveals Release Effects and an Ontogenetic Shift"

_animals, 2023, doi:10.3390/ani13071218_

Round 1
Reviewer 1 Report
See attached for specific comments and suggestions. Overall, the manuscript is focused on the dispersal of released turtles but does not adequately address their potential recruitment to local foraging grounds. Over half the turtles released in January appeared to recruit to nearshore waters but there was no mention of the age classes for these residential turtles, only those that dispersed (line 283). Do younger turtles tend to disperse and older turtles become residential? This would fit the life history model for a shift from pelagic to neritic habitats. Need to provide more information on the potential neritic foraging habitats around the Caymans (fishery data of Bell et al., 2006) and the aggregations of turtles inhabiting these areas (such as Blumenthal et al., 2010).

Author Response
We thank the reviewer for their helpful comments and suggestions, especially on incorporating more discussion of the “residential” turtles released in January that remained near the Island for the duration of their tags. We have worked to address these suggestions throughout the text.
For all more minor suggested edits, we have incorporated these into the revised version of the manuscript (please see revised manuscript).
For other suggested larger edits and changes, we reference original line numbers of these comments, our responses, and new line numbers with requested changes:
- LL203: We have added description and results for circular statistical tests (Watson’s U2, using the R package “circular”) for comparing turtle daily headings with mean daily current directions, as well as comparing turtle daily headings between tracking periods (January vs July). Results of all tests showed significant (P-value < 0.001) for all groups.
LL218-220: “Statistical differences between mean turtle headings (January vs. July) and mean ocean current directions were further examined using a Watson’s U2 two sample test with the “circular” R package (Agostinelli and Lund 2022).”
LL325-338: ” Circular histograms and statistical tests of daily magnetic headings for turtles that dispersed from the island (“dispersive” turtles) showed significant differences in mean headings between turtles released in January vs. July (Watson’s U2 two sample test, t-value: 0.620, p-value: < 0.001). The majority of turtles released in January had a mean south-west heading throughout their dispersal, whereas those released in July had a mean north-east heading (January: 226 deg ± 101 SD, July: 20 deg ± 94 SD; Fig 1.B). Significant differences were further seen in between the mean headings of turtles and the mean direction of ocean currents for each tracking period (Watson’s U2 two sample test, January t-value: 0.703, July t-value: 1.101, January and July p-values: < 0.001) (Fig 1.B). Mean current directions experienced by turtles along their tracks were also significantly different between tracking periods (Watson’s U2 two sample test, t-value: 0.402, p-value: < 0.001), with currents for July turtles being more north-westerly than currents for January turtles (January currents: 288 deg ± 76 SD, July currents: 331 deg ± 75 SD; Fig 1.B).”
- LL282, LL494, LL502: We have added an additional column to Table S1 in the Supplement, annotating for each individual whether it was “dispersive” (left the island) or “residential” (remained near the island) for easy visual assessment of the age classes for these individuals. There was actually a similar number of turtles represented for the 1-2 and 2-3 yr age classes for “dispersive” vs “residential” turtles for those released in January (5 1-2 yr olds, and 4 2-3 yr olds, for both “dispersive” and “residential” January turtles); but there was a higher number of turtles in the 3-4 yr age class for “residential” turtles in January compared to “dispersive” (7 “residential” vs 3 “dispersive” 3-4 yr olds). July turtles were all one age class (1-2 yrs) and all dispersive, which as discussed in the text (LL465-495), we believe may be more of an artifact of the release strategy, where turtles were released into favorable currents directing them away from the island. Due to the shortness of the tracks (a couple had tracks of only 3 days in duration) and the small sample sizes, it is difficult to determine if the higher number of “residential” 3-4 yr olds in January is due to their habitat preferences at this size and age or other variables (as discussed in the text LL422-425, they may have eventually dispersed from the island but it was unable to be recorded due to short-lived tag lives or tag loss). However, the points the reviewer has brought up about the potential long term recruitment of these individuals to local foraging habitats is an important one (thank you!) and we have added some text to the manuscript describing the number of residential individuals in each age class and discussion of this:
LL300-306: “Of the 22 turtles that dispersed from the island in both January and July (“dispersive” individuals), there was an uneven distribution in the number of individuals represented from each age class (N=15 1-2 year olds, N=4 2-3 year olds, and N=3 3-4 year olds in January, N=10 1-2 year olds in July) (Table 1). Of the remaining 16 turtles that stayed within the vicinity (10 km) of Grand Cayman Island for the duration of their tags ("residential”), there was also an uneven distribution in the age classes (N=5 1-2 year olds, N=4 2-3 year olds, and N=7 3-4 year olds) (Table S1).”
LL425-443: “Another possibility, although additionally impossible to determine due to the shortness in tag durations, is that some or all of these turtles remaining near the island took up residential behavior and recruited to local habitats. All of the “residential” turtles that remained near the island were of body sizes that have been recorded in another study (mean: 55.7 cm CCL, range: 32.8 – 80.7 cm CCL; Blumenthal et al 2010) to inhabit a local southern lagoon, where diverse behavior (e.g., moving in and out of the lagoon, diurnal seagrass and coral reef use) was observed. Differences seen between the average number of positions reported per day for “dispersive” versus these “residential” turtles in our study, with far more positions reported per day on average for “dispersive” turtles (mean: 12.6 positions per day) compared to “residential” turtles (mean: 4.2 positions per day), further supports that the 16 turtles remaining near the island for the duration of their tags may have been performing residential behaviors, such as less frequent diving and more benthic resting or foraging behavior. Interestingly, there was a higher number of individuals within the larger age class of our study (3-4 year olds, 54-62 cm CCL, Table S1) for these “residential” turtles (N=7 3-4 year olds) versus for the “dispersive” turtles (N=3 3-4 year olds). These older and larger juveniles (close to the mean size of 55.7 cm CCL observed by Blumenthal et al. 2010) may have been more likely to recruit locally due to being closer in size and age to when the lifestage shift between pelagic to neritic developmental habitat occurs (Moncada et al 2006, Reich et al 2007, 496 Meylan et al 2011).”
- LL383: For the two turtles that recorded a single location, the reported locations likely cannot be trusted. One of these individuals (ID 214082) had its one location reported before the start date of the study, which invalidates it. The other individual (ID 203086) had its one location far away from the island (Lon -84.4, Lat 16.8) and is thus unusable, as there is no way of determining if this was a valid position or not. We have updated the text here with the correct number (N=28) after removing these two individuals.
- LL386, LL404: It is certainly possible that the “residential” turtles may have remained near the Island and fully recruited there, and all turtles recorded as “residential” were within the size ranges documented by Blumenthal et al (2010). We thank the reviewer for their suggestions and have added text to the Discussion on the potential recruitment of “residential” turtles to the Islands:
LL425-443: ” Another possibility, although additionally impossible to determine due to the shortness in tag durations, is that some or all of these turtles remaining near the island took up residential behavior and recruited to local habitats. All of the “residential” turtles that remained near the island were of body sizes that have been recorded in another study (mean: 55.7 cm CCL, range: 32.8 – 80.7 cm CCL; Blumenthal et al 2010) to inhabit a local southern lagoon, where diverse behavior (e.g., moving in and out of the lagoon, diurnal seagrass and coral reef use) was observed. Differences seen between the average number of positions reported per day for “dispersive” versus these “residential” turtles in our study, with far more positions reported per day on average for “dispersive” turtles (mean: 12.6 positions per day) compared to “residential” turtles (mean: 4.2 positions per day), further supports that the 16 turtles remaining near the island for the duration of their tags may have been performing residential behaviors, such as less frequent diving and more benthic resting or foraging behavior. Interestingly, there was a higher number of individuals within the larger age class of our study (3-4 year olds, 54-62 cm CCL, Table S1) for these “residential” turtles (N=7 3-4 year olds) versus for the “dispersive” turtles (N=3 3-4 year olds). These older and larger juveniles (close to the mean size of 55.7 cm CCL observed by Blumenthal et al. 2010) may have been more likely to recruit locally due to being closer in size and age to when the lifestage shift between pelagic to neritic developmental habitat occurs (Moncada et al 2006, Reich et al 2007, 496 Meylan et al 2011).”
LL553-565: ” Despite the resolution of our data potentially restricting our ability to detect a relationship between Sargassum density and turtle swimming persistence, floating patches of Sargassum algae are considered a key foraging resource for juvenile green turtles in the pelagic stage (Witherington et al 2012) and it is possible that juveniles foraged or associated with these patches during dispersal. Additionally, as previously discussed (see section 4.1, Effects of Release Location and Timing on Dispersal), the majority of turtles released in January remained near the vicinity of Grand Cayman island for the duration of their tags (N=16) and were all within the size range for other juvenile greens (mean: 55.7 cm CCL, range: 32.8 – 80.7 cm CCL; Blumenthal et al 2010) that have been previously recorded to inhabit local nearshore lagoons, sea grass beds, and coral reefs (Blumenthal et al 2010). Further research using tags with longer tracking durations and possibly combined with passive acoustic arrays or mark-recapture studies (Bluementhal et al 2010) is needed to determine if head-started juvenile greens are recruiting to local foraging and resting habitats of the Cayman Islands.”
LL608-612: “Further, although most individuals released in January seemed to follow ocean cur-rents to return and potentially stay near the vicinity of Grand Cayman island for the duration of their tags, as previously discussed, there are known habitats capable of supporting development of juvenile greens around the Islands (Blumenthal et al 2010) and their residency may help support recruitment long-term to the local nesting population of Caymanian green turtles, which has been estimated to have only 100-150 nesting females (Blumenthal et al 2021).”
- LL 527: We thank the reviewer for providing the Shaver et al (2016) resource on head-starting for Kemps ridley turtles. This resource specifically discusses how head-starting might have affected turtle navigational abilities, natural imprinting, and nesting. We have incorporated this resource within the text.
LL521-526: “For turtles in July, four individuals were recorded to essentially return “home” (back to the vicinity of the Cayman Islands) by the end of their tracks (IDs 229669, 229672, 229676, 229677), broaching the possibility of homing or imprinting on the rearing location for captive-raised and/or juvenile turtles (Shaver et al 2016), and indeed, all four of these individuals appeared to have swimming vectors directed towards the Islands towards the end of their tracks (Fig 2.B).”
LL588-600: “Head-starting and release initiatives, where animals are born or hatched in captivity, raised to an age where their larger body sizes make natural mortality less likely, and released into the wild, have often been deemed unsuccessful ways of increasing sea turtle survivorship (Frazer 1992, Heppell et al 1996) and potentially disruptive to turtles’ natural navigational abilities, natal imprinting, and nesting site choices (Shaver et al 2016). However, these criticisms may largely be due to a lack of monitoring and/or tracking efforts for most head-starting projects that can support evidence of long-term success (Burke 2015); additionally, these studies may not account for the effects of oceanography (e.g., ocean currents) on dispersal. As our results show, release location and timing with respect to ocean current directions have potentially important implications for where released, head-started turtles end up, with January individuals in our study following the direction of ocean currents back to their rearing location but July turtles following these ocean currents away from their rearing location.”
- LL337: No, flipper and PTT tags were unfortunately not used in this study.
Reviewer 2 Report
This is interesting research about the green turtles in the Caribbean Sea that will allow us to understand more about their movements, and behavior.

Author Response
We thank the reviewer for their suggestions and comments. We have worked to address these issues throughout the manuscript.
For all minor suggested edits, we have incorporated these in text (please see revised manuscript).
For all larger suggested edits or questions, we respond below with the original line numbers of the comments, our response, and line numbers of any new changes:
- LL23: Yes, the ages referenced in the abstract (1-2, 2-3, 3-4 years) is the captivity time. As discussed further in the text, these individuals were all head-started (hatched and raised in captivity until release to improve survivorship). We have added the “head-started” term here to indicate this.
- LL65: We are referencing multiple species here (e.g., see the cited Hazen et al 2012 paper).
- LL94: To our knowledge, and as stated, simultaneous tracking of multiple age classes has rarely been performed for juvenile sea turtles and there are large knowledge gaps on mechanisms and patterns of dispersal for many species and populations. Studies on dispersal for one particular age/size class, species, and region may not be a good descriptor of behavior for another species and region of interest. Although there have been studies testing in laboratories to determine drivers of dispersal for neonates (e.g., magnetic fields, Lohmann et al 2004, Fuxjager et al 2011), simulations to determine the role of active versus passive dispersal with respect to current drift (e.g., Gaspar et al 2012, Putman et al 2012), and some empirical tracking studies of juveniles for some species and regions (e.g., mostly loggerheads but some greens, Mansfield et al 2012, 2017, 2021), empirical tracking studies of juvenile turtles are still rare, especially for multiple age classes at a time. Questions remain for many populations, for example when the shift from pelagic to neritic developmental habitat occurs and drivers of dispersal patterns (ocean currents? magnetic field? habitat availability?).
- LL101-104: We thank the reviewer for their comment but here reply that this particular sentence does already describe and reference studies done on juvenile green turtles. Are there other additional studies that the reviewer would suggest we incorporate here?
- LL124-127: We thank the reviewer for their comment but we believe this sentence is essential for the Introduction, as it states the larger goal for and justification of the analysis.
- LL140: We thank the reviewer for their comment and have added a map of the study area to the Supplement and referenced it in the Methods (see Figure S1 in Supplement).
- LL145: The Health and Screening Protocol used by the Cayman Turtle Conservation and Education Centre (CTCEC) is not available publicly but is available via request to the CTCEC . We have added a statement about this here.
- LL184: Yes, the information stated on this line is correct. Please see the included reference for this product for further detail.
- LL199: We are unsure what the reviewer means here by their statement “What happens with the other two turtles?”; this particular sentence is describing how initial locations were used to determine the initial headings of the turtles. All other locations were used to determine daily headings for turtles (see LL202).
- LL256: For the two turtles that only had one location reported, these locations cannot be trusted or used to infer where the turtles went. One of these individuals (ID 214082) had its one location reported before the start date of the study, which invalidates it. The other individual (ID 203086) had its one location far away from the island (Lon -84.4, Lat 16.8) and is thus unusable, as there is no way of determining if this was a valid position or not.
- LL264, LL494: Here, we would reference the reviewer to the Supplement, which contains a larger Table with information on turtle body sizes (Table S1). We have added body size information and measurements taken additionally to the methods text as suggested (LL164-165).
- LL347: We thank the reviewer for their helpful suggestion and have added the total displacement (distance, in km, from start locations near the Cayman Islands to final locations at the end of turtle tracks) to Table 1 (see column, Total Displacement).
Reviewer 3 Report
It is definitely something innovative to answer how hatchlings behave and disperse after reaching the ocean, which is an already difficult topic to study and have definite answers to.
The manuscript needs just a few minor corrections, mostly to adjust to the styling guidelines of the journal. My suggestion to the authors, read carefully the author's instructions provided by the journal.
Also, this reference Wildermann N, Critchell K, Fuentes MMPB, Limpus CJ, Wolanski E, Hamann M 2017 Does behaviour affect the dispersal of flatback post-hatchlings in the Great Barrier Reef? R. Soc. open sci. 4: 170164. http://dx.doi.org/10.1098/rsos.170164 might be an interesting reading to add or have in mind for future research.
Some comments and suggestions are in the attached file.

Author Response
We thank the reviewer for their suggestions, we have incorporated all minor suggestions throughout the text (please see revised manuscript).
Additionally, we have revised the text as suggested by: shortening the abstract to the maximum allowed length of 200 words, adding the Conclusion section into the Discussion, updating the references to be numbered in order and cited according to the instructions, and we have updated the “Back Matter” to follow the author guidelines.
For Figure 1, we have created a more high resolution figure and better centered the plots to make them more visually appealing. However, the massive difference in scale of areas used by turtles released in January versus July unfortunately means that we still believe using maps with two different scales is best, as a “zoomed out” map for July turtles would result in tracks not being well visualized against the ocean currents (they would look just like a colored clump due to the smaller area used and clustering in movement in July).
As to the question about why we used a “different sample size of turtles” each time, unfortunately we were limited by turtle availability and could only get the youngest age class (1-2 years) in July. However, despite having different overall sample sizes (January: N=30, July: N=10), we did have equal sample sizes within each age class during each time period to allow for direct comparisons (January: 10 1-2 yr olds, 10 2-3 yr olds, 10 3-4 yr olds, July: 10 1-2 yr olds).
We additionally thank the reviewer for sharing the Wildermann et al (2017) paper and have incorporated this a reference in the Introduction.
Reviewer 4 Report
This manuscript by Barbour et al. represents a well-designed and informative work. The researchers performed a large-scale experiment on captive-reared green turtle juveniles to shed light on their behavior and ontogeny post-release. The work found specific factors (release time and location) to be seemingly more significant compared to other environmental drivers (sargassum, sea temperature), but given technological limitation, success of captive release could not be determined. I deem this study an important contribution to conservation efforts of endangered Caribbean green turtles. I have noticed some minor oversights in grammar and structure, lack of discussion of certain findings that could be useful, and slightly confusing information placement that needs replacing across the sections. The manuscript could also benefit from providing more info on the rearing practice of the green turtles under study. This is all indicated with comments throughout the document. It is unfortunate that tag technology is still incredibly short-term compared to what is needed to determine a successful release, an average tracking time of 15 days is not nearly enough to understand long-term adaptation and survival patterns of captive-reared turtles, however this is repeatedly and clearly outlined by the authors and encouragement to more fine scale studies is highly appropriate. I hope the authors will find my comments beneficial to their manuscript.

Author Response
We thank the reviewer for their helpful review and suggestions. We have incorporated all minor edits indicated (please see revised manuscript).
Additionally, we have addressed larger suggestions and questions throughout the manuscript and respond about these below:
- LL90: We thank the reviewer for their suggestion and have added a short statement here about the progression of satellite tags: “The progression and miniaturization of long-lasting satellite tags (now on the order of 10 g weight or less and capable of recording tracks over 100 days in duration) has recently allowed for empirical measurements of juvenile sea turtle movement (Mansfield et al 2012, Kobayashi et al 2014, Putman and Mansfield 2015, Briscoe et al 2016, Mansfield et al 2017).”
- LL130: We have added a short definition of persistence velocity to clarify this term: “A generalized additive mixed model (GAMM) of turtle persistence swimming velocity (or the tendency of turtles to swim persistently in a given direction, km/day)…”
- LL143: We thank the reviewer for their suggestion and have added details as to the captive rearing practices:
LL132-141: “The turtles were hatched and raised in a captive setting, being reared in various sized concrete tanks (up to 150 m2, roughly I meter depth, supplied with direct, unfiltered seawater), fed a nutritionally complete, floating and modified extruded fish diet, and being kept in groups of like size and age until they had reached the desired age class for release (further details on captive rearing can be found in the “Commercial Reptile Farming” section of Mader's Reptile Medicine and Surgery, 36). All turtles were quarantined prior to release and were checked to ensure that they each fulfilled requisite health and quarantine requirements (according to CTCEC Quarantine and Health Screening Protocol, available upon request to the CTCEC) for release.”
- LL150: We have added some short test justifying the use of two different tag types: “Turtles were either tagged with miniature (~1 g) solar (N=25) or battery-powered satellite tags (N=15); the use of these different tags allowed for comparison of tag performance between the two tag types but was also due to tag availability.”
- LL172: We thank the reviewer for their suggestion and have added definitions of “dispersive” vs “residential” turtles earlier in the text (section 2.1).
LL175-177: ” Turtles were categorized as being either “dispersive”, if their recorded tracks left the vicinity (~10 km) of Grand Cayman island, or “residential”, if their tracks remained in the vicinity of Grand Cayman island after release.”
- LL198: We have added to the text here the temporal duration of the initial recorded locations used to create the histograms: “…circular histograms were used on the first two recorded locations for each turtle with viable locations (N=38; January: mean of 25 hrs ± 53 SD, July: mean of 18 hrs ± 10 SD).”
- LL258: We have added the approximate number of days it took turtles to return to the island after release in January: ”Of the remaining turtles released in January with viable positions (N=28), upon release, all moved in a primarily southerly direction back towards Grand Cayman island, generally matching the direction of the surface drifter (Fig 1.A) and returning to the vicinity of the Island within 24 hours of release.”
- LL290: We thank the reviewer for their comment. We have added some text to the Discussion about this significant result. Additionally, we have added some text about why dispersive turtles may have reported more positions per day than residential turtles: this may have been due to the fact that “dispersive” turtles were demonstrating traveling behavior with frequent dives and surfaces, in comparison to “residential” turtles that may have been demonstrating more benthic foraging or resting behavior, with more time spent on the bottom and less frequent diving/surfacing. Both residential and dispersive turtles had a mixture of solar and battery tags (see Table S1, in the Supplement).
LL644-648: ”Tags overall reported large amounts of error in positions (especially for January turtles, Fig S4), with battery-powered tags having a significantly lower number of positions reported compared to solar-powered tags (Fig S2.C). Future studies should work with modified versions of these tags, prioritizing solar-powered ones based on these results.”
LL425-443: ”Another possibility, although additionally impossible to determine due to the shortness in tag durations, is that some or all of these turtles remaining near the island took up residential behavior and recruited to local habitats. All of the “residential” turtles that remained near the island were of body sizes that have been recorded in another study (mean: 55.7 cm CCL, range: 32.8 – 80.7 cm CCL; Blumenthal et al 2010) to inhabit a local southern lagoon, where diverse behavior (e.g., moving in and out of the lagoon, diurnal seagrass and coral reef use) was observed. Differences seen between the average number of positions reported per day for “dispersive” versus these “residential” turtles in our study, with far more positions reported per day on average for “dispersive” turtles (mean: 12.6 positions per day) compared to “residential” turtles (mean: 4.2 positions per day), further supports that the 16 turtles remaining near the island for the duration of their tags may have been performing residential behaviors, such as less frequent diving and more benthic resting or foraging behavior. Interestingly, there was a higher number of individuals within the larger age class of our study (3-4 year olds, 54-62 cm CCL, Table S1) for these “residential” turtles (N=7 3-4 year olds) versus for the “dispersive” turtles (N=3 3-4 year olds). These older and larger juveniles (close to the mean size of 55.7 cm CCL observed by Blumenthal et al. 2010) may have been more likely to recruit locally due to being closer in size and age to when the lifestage shift between pelagic to neritic developmental habitat occurs (Moncada et al 2006, Reich et al 2007, Meylan et al 2011).”
- LL303: We have added the proportion of individuals following the observed mean headings for January and July to the text: “A large proportion (42% or 5/12 turtles) of turtles released in January had a mean south-west heading throughout their dispersal, whereas a large proportion (50% or 5/10 turtles) of those released in July had a mean north-east heading (January: 226 deg ± 101 SD, July: 20 deg ± 94 SD; Fig 1.B).”
- LL362: We thank the reviewer for their comment here, we have softened the language to insinuate that based on the observed tracks, the turtles in our study have potential to recruit to nearshore, Caribbean habitats (including at the Cayman Islands). However, the shortness in our track durations prevented confirmation of long-term recruitment and we have discussion of this in the text, with additional sources from other studies observing juvenile greens in these habitats (LL556-573). Additionally, based on another reviewer’s comment, we have incorporated more discussion of the “residential” turtles from our study (turtles that stayed around the vicinity of the Island for their tracks), which seemed to have strong potential for long-term recruitment and made up the majority of tagged individuals released in January (18/28).
LL394-400: “Many of the individuals in our study demonstrated an ontogenetic niche shift, either swimming with persistence through pelagic waters towards nearshore areas around the Caribbean (“dispersive” turtles) or remaining nearshore of the Cayman Islands for the duration of their tracks ("residential” turtles). This supports that with longer tracking durations, turtles would likely have been observed to recruit long-term to a variety of nearshore areas known to serve as developmental habitat around the Caribbean and the Cayman Islands.”
LL425-443:” Another possibility, although additionally impossible to determine due to the shortness in tag durations, is that some or all of these turtles remaining near the island took up residential behavior and recruited to local habitats. All of the “residential” turtles that remained near the island were of body sizes that have been recorded in another study (mean: 55.7 cm CCL, range: 32.8 – 80.7 cm CCL; Blumenthal et al 2010) to inhabit a local southern lagoon, where diverse behavior (e.g., moving in and out of the lagoon, diurnal seagrass and coral reef use) was observed. Differences seen between the average number of positions reported per day for “dispersive” versus these “residential” turtles in our study, with far more positions reported per day on average for “dispersive” turtles (mean: 12.6 positions per day) compared to “residential” turtles (mean: 4.2 positions per day), further supports that the 16 turtles remaining near the island for the duration of their tags may have been performing residential behaviors, such as less frequent diving and more benthic resting or foraging behavior. Interestingly, there was a higher number of individuals within the larger age class of our study (3-4 year olds, 54-62 cm CCL, Table S1) for these “residential” turtles (N=7 3-4 year olds) versus for the “dispersive” turtles (N=3 3-4 year olds). These older and larger juveniles (close to the mean size of 55.7 cm CCL observed by Blumenthal et al. 2010) may have been more likely to recruit locally due to being closer in size and age to when the lifestage shift between pelagic to neritic developmental habitat occurs (Moncada et al 2006, Reich et al 2007, 496 Meylan et al 2011).”
LL553-565: ” Despite the resolution of our data potentially restricting our ability to detect a relationship between Sargassum density and turtle swimming persistence, floating patches of Sargassum algae are considered a key foraging resource for juvenile green turtles in the pelagic stage (Witherington et al 2012) and it is possible that juveniles foraged or associated with these patches during dispersal. Additionally, as previously discussed (see section 4.1, Effects of Release Location and Timing on Dispersal), the majority of turtles released in January remained near the vicinity of Grand Cayman island for the duration of their tags (N=16) and were all within the size range for other juvenile greens (mean: 55.7 cm CCL, range: 32.8 – 80.7 cm CCL; Blumenthal et al 2010) that have been previously recorded to inhabit local nearshore lagoons, sea grass beds, and coral reefs (Blumenthal et al 2010). Further research using tags with longer tracking durations and possibly combined with passive acoustic arrays or mark-recapture studies (Bluementhal et al 2010) is needed to determine if head-started juvenile greens are recruiting to local foraging and resting habitats of the Cayman Islands.”
- LL382: We thank the reviewer for catching this error, it should read N=28 (two out of the 30 had unviable positions).
- LL399: We have clarified in the text but also here, that the July turtles dispersed immediately (there was no delay in dispersal, unlike the January turtles) upon release from the Island.
- LL468: We thank the reviewer for their suggestion and have added text about these turtles being within the smaller age class. It is difficult to determine if this is a factor of their age class, as turtles released in January of the same age class (in fact, with somewhat smaller body sizes, Table S1) did not demonstrate this behavior. As described in the text (LL531-540), this could be more of “confused” behavior due their “hard” release strategy compared to January turtles. We have added some text discussing this:
LL521-530: “For turtles in July, four individuals were recorded to essentially return “home” (back to the vicinity of the Cayman Islands) by the end of their tracks (IDs 229669, 229672, 229676, 229677), broaching the possibility of homing or imprinting on the rearing location for captive-raised and/or juvenile turtles (Shaver et al 2016), and indeed, all four of these individuals appeared to have swimming vectors directed towards the Islands towards the end of their tracks (Fig 2.B). These four individuals were within the youngest age class (1-2 years); however, as none of the 1-2 year old individuals released in January demonstrated this possible homing behavior, it may be more of a signal of the release strategy and current structures around the Island in July, rather than the size or age of the turtles.”